# Determination of time-varying periodicities in unequally spaced time series of OH* temperatures using a moving Lomb-Scargle Periodogram and a fast calculation of the false alarm probabilities

Christoph Kalicinsky[1], Robert Reisch[1], Peter Knieling[1], and Ralf Koppmann[1]

[1]Institute for Atmospheric and Environmental Research, University of Wuppertal, Germany

**Correspondence:** C. Kalicinsky (kalicins@uni-wuppertal.de)

**Abstract.** We present an approach to analyse time series with unequal spacing. The approach enables the identification of significant periodic fluctuations and the derivation of time-resolved periods and amplitudes of these fluctuations. It is based on the classical Lomb-Scargle periodogram (LSP), a method that can handle unequally spaced time series. Here, we additionally use the idea of a moving window. The significance of the results is analysed with the typically used false alarm probability (FAP). We derived the dependencies of the FAP levels on different parameters that either can be changed manually (length of the analysed time interval, frequency range) or that change naturally (number of data gaps). By means of these dependencies we found a fast and easy way to calculate FAP levels for different configurations of these parameters without the need of a large number of simulations. The general performance of the approach is tested with different artificially generated time series and the results are very promising. Finally, we present results for nightly mean OH* temperatures that have been observed from Wuppertal (51° N, 7° E, Germany).

*Copyright statement.* TEXT

## 1 Introduction

Many time series in atmospheric sciences are characterised by an unequal spacing of the data points, e.g. due to data gaps. OH or other airglow observations often have such data gaps in the measured time series (e.g. Espy et al., 1997; Das and Sinha, 2008; Reid et al., 2014). The OH* temperatures which have been observed from Wuppertal (51° N, 7° E) since the 1980s also exhibit an unequal spacing. The time series of nightly mean OH* temperatures repeatedly has data gaps mainly because of bad weather conditions during some nights that prevent useful measurements (see e.g. Bittner et al., 2000). Within a single night such data gaps can also occur when clouds move through the line of sight. The measurements before and after such a cloud contimination are useful. Typical methods such as the fast Fourier transformation (FFT) rely on a discrete sampling with equal spacing. Thus, a time series like that of OH* temperatures has to be manipulated, e.g. with interpolation techniques before the analysis (e.g. Espy et al., 1997; Bittner et al., 2000; Reid et al., 2014). The Lomb-Scargle Periodogram (LSP; Lomb, 1976; Scargle, 1982) is a method that can handle this drawback, as it can be used for time series with unequal spacing. This method

has been used in different studies analysing airglow observations (e.g. Espy et al., 1997; Takahashi et al., 2002; Gao et al., 2010; Reid et al., 2014; Egito et al., 2018; Franzen et al., 2018; Nyassor et al., 2018).

A second important point with respect to the analysis of periodicities is the variation of these periodicities with time, i.e. the period is not stable during the complete analysed time interval or the amplitude varies. In such cases many methods as the FFT and the LSP will lead to results of a mean state only. The wavelet transform is a method that is very useful then as it delivers time-resolved information on the periodicities of the analysed time series and it is used in several studies analysing the temporal evolution of periodic signals in airglow observations (e.g. Das and Sinha, 2008; Takahashi et al., 2013; Reid

et al., 2014; Nyassor et al., 2018). In the case of the Wuppertal OH* temperatures Bittner et al. (2000) used the wavelet to analyse the variability of the nightly mean OH* temperatures after assimilation of the data gaps in the time series by use of the maximum entropy method (MEM). Similar to that other studies also report that the time series have to be interpolated before the use of the wavelet transform (e.g. Das and Sinha, 2008; Reid et al., 2014) or the sampling is at least almost evenly distributed (Nyassor et al., 2018). The goal of the presented study is to avoid such an assimilation of the data gaps and still

derive time-resolved information on the periodicities. Thus, we combined the LSP and the idea of a moving window to identify and characterise periodicities in unequally spaced time series even when the periodicities vary with time. Other airglow studies also use some kind of windowed LSP, but for independent time windows following each other such as different parts of a night (Reid et al., 2014) or months of a year (Egito et al., 2018). Some studies analysing radar observations of winds report of periodogram analysis with a moving window (Yoshida et al., 1999, but without significance evaluation) or a LSP analysis

for at least partly overlapping windows (Luo et al., 2000). However, our study combines the LSP with a moving window (moved with the minimum possible time step) and, additionally, we derive a fast and easy method to calculate the false alarm probabilities (FAP) for different situations (length of time series, frequency range, data gaps) to identify significant results. The determination of the FAP levels is typically done with Monte-Carlo simulations which is very time-consuming (e.g. Cumming et al., 1999; Zechmeister and Kürster, 2009). Thus, our new empirical derived relationship to calculate the levels improves the

application of the method.

The main intention of the paper is to describe the approach from a user perspective and to illustrate the capabilities of the approach with examples of artificial data sets as well as observations. The paper is structured as follows. In Sect. 2 the classical LSP and the new approach are explained. The evaluation of the significance of obtained results is made in Sect. 3. Finally, the method is applied to artificial data and observations of OH* temperatures in Sect. 4. A short summary is given in Sect. 5.

## 50  2  Methodology

### 2.1  Classical Lomb-Scargle Periodogram

The Lomb-Scargle Periodogram (LSP) was developed by Lomb (1976) and Scargle (1982). The periodogram is defined as

$$P_X(\omega) = \frac{1}{2} \left\{ \frac{\left[ \sum_j X_j \cos \omega (t_j - \tau) \right]^2}{\sum_j \cos^2 \omega (t_j - \tau)} + \frac{\left[ \sum_j X_j \sin \omega (t_j - \tau) \right]^2}{\sum_j \sin^2 \omega (t_j - \tau)} \right\}, \tag{1}$$

where $X_j$ are the measurements at the times $t_j$, $\omega$ is the angular frequency ($\omega = 2\pi f$), and the time offset $\tau$ is defined as

$$\tan(2\omega\tau) = \frac{\left( \sum_j \sin 2\omega t_j \right)}{\left( \sum_j \cos 2\omega t_j \right)}. \tag{2}$$

An advantage compared to other methods such as the FFT is that the LSP can handle unequally spaced time series. A prerequisite is that the time series has zero mean before the calculation of the periodogram powers. With the given definition the LSP has two useful properties: 1) It is invariant to a shift of the origin of time. 2) It is equivalent to the least squares fitting of sinusoids (e.g. Horne and Baliunas, 1986). Scargle (1982) showed that the definition of the periodogram is the same (except 60  for a factor of 1/2) as the reduction in sum of squares (sum of squares of data – sum of squares of residual) when using least squares fitting of sinusoids (see Scargle, 1982, Appendix C). Thus, the maximum power in the periodogram occurs at that frequency that leads to a minimum of the sum of squares of the residuals when a sinusoid with this frequency is fitted to the time series.

### 2.2  Moving LSP

The aproach used in the following analyses is based on the classical LSP, but the whole time series is analysed sequentially. The procedure is as follows:

A window size (time interval) , which is typically much smaller than the length of the whole time series, is defined. Then the procedure starts at the beginning of the time series.

1. the LSP for the data points within the window (time interval) is calculated

2. the window is moved by 1 time step (minimum possible sampling step)

3. move to step one until the end of the times series is reached

By executing this procedure one single LSP is calculated for each possible part of the time series with the length of the window (time interval). By contrast to the LSP for the whole time series at once, this procedure delivers time-resolved information on the periodicities and amplitudes.

## 2.3 Normalisation of the LSP

There are different ways to normalise the periodogram: sample variance (or sum of squares), known variance of data, and variance of the residuals (see e.g. Cumming et al., 1999; Zechmeister and Kürster, 2009). Here we use the normalisation by the sample variance and sum of squares, respectively. These two only differ by a constant factor that relies on the number of data points $N$. The periodogram power can vary between 0 and $(N-1)/2$ when using the normalisation by the sample variance and between 0 and 1 when using the normalisation by sum of squares (when the factor 1/2 is also considered (compare Sect. 2.1)) (e.g. Cumming et al., 1999; Zechmeister and Kürster, 2009). As the height of a peak in the case of the normalisation by the variance depends on the number of data points N, the peak heights for the same sinusoid differ for different number of data points. Since the data gaps in the time series of nightly mean OH* temperatures are randomly distributed, the number of data points in different possible windows of same size can vary. In order to make the peak heights in these different windows comparable we prefer the normalisation by the sum of squares. This type of normalisation has another useful property. Because of the equivalence to the reduction in sum of squares when fitting a sinusoid, the normalisation by the sum of squares leads to a normalised power that gives the contribution of the sinusoid to the total sum of squares, and, therefore, to the total variance. In this way it is a measure of the explained variance. Here uncorrelation between different sinusoids and/or a sinusoid and the residual is assumed. This is, at least approximately (increasing with number of data points), the case for sinusoids with different periods and, thus, the variances of the individual parts (sinusoids) of the time series add up.

Alternatively, one can determine the amplitude of the sinusoid at each frequency. This is also based on the equivalence of the periodogram power and the reduction in sum of squares. Furthermore, the variance of a sinusoid is given by $A^2/2$, where $A$ is the amplitude (e.g. Horne and Baliunas, 1986; Smith, 1997). With these two relationships the amplitude can be calculated as

$$A(\omega) = \sqrt{\frac{4P_X(\omega)}{N-1}}. \tag{3}$$

In total the LSP delivers information on the periodicities together with a measure of the explained variance when a sinusoid is fitted to the data and the corresponding amplitude of the sinusoid. An example periodogram is shown in Fig. 1. The time series that is analysed is a combination of two sinusoids with different periods and amplitudes. The first one has a period of 10 days and an amplitude of 1 K whereas the second sinusoid has a period of 35 days and an amplitude of 0.5 K. The total length of the time series is 60 days and the time series has equal spacing. Thus, the variance of the second sinusoid is only one quarter of the variance of the first one. This can be seen in the normalised power (black curve in Fig. 1) where at 10 days a value of about 0.8 is reached and at 35 days a value of about 0.2. Because of the different amplitudes, the sinusoids contribute to 80 and 20 % to the total variance of the time series, respectively. And also the amplitudes themselves are well determined by using Eq. 3 (see red curve in Fig. 1).

## 3 Significance evaluation

 ### 3.1 False alarm probability

An important quantity with respect to the LSP is the so called false alarm probability (FAP). It gives the probability that a peak with a height above a certain level can occur just by chance, e.g. due to noise. The distribution of the periodogram powers and, thus, the description of the false alarm probability depends on the type of normalisation (see e.g. Cumming et al., 1999; Zechmeister and Kürster, 2009). In the case of the normalisation by the sample variance the periodogram powers follow a beta distribution (Schwarzenberg-Czerny, 1998). As the variance and the sum of squares differ by a constant factor only, the type of distribution is the same. Hereafter, we only describe the situation for the normalisation by sum of squares. At a single frequency the probability that a peak height $z$ exceeds a value of $z_0$ is given by

$$\text{Prob}(z > z_0) = (1 - z_0)^{\frac{N-3}{2}}, \tag{4}$$

where $N$ is the number of data points (Zechmeister and Kürster, 2009). Since periods in a frequency range are analysed, one is interested in the probability that one peak somewhere in the periodogram covering a frequency range $\Delta f$ exceeds a certain value by chance, which is given by the FAP. The probalitity that all peaks in this frequency range are below or equal or certain value is given by $(1 - \text{Prob}(z > z_0))^{N_i}$, where $N_i$ is the number of independent frequencies (number of frequencies where potentially peaks can occur). Then the FAP is

$$\text{FAP} = 1 - (1 - \text{Prob}(z > z_0))^{N_i}, \tag{5}$$

where $N_i$ gives the number of independent frequencies (see e.g. Horne and Baliunas, 1986; Cumming et al., 1999; Zechmeister and Kürster, 2009, for some discussion on FAP). There is no analytical way to describe the number of independent frequencies, but a good way to determine $N_i$ is the use of Monte-Carlo simulations (see e.g. Cumming et al., 1999).

The procedure to determine $N_i$ using simulations is as follows. As already pointed out by Scargle (1982) the cumulative distribution function (CDF) can be used to determine the FAP. We use a large number of samples of random values taken from a Gaussian distribution each. Then we calculate the LSP for each sample and determine the height of the maximum peak within the analysed frequency range. From these maximum peak heights we calculate the empirical CDF which gives the probability that the maximum peak and thus all other peaks in a periodogram have a height equal or below a certain value. The CDF is then given by $(1 - \text{Prob}(z > z_0))^{N_i}$ and consequently, the FAP is then 1-CDF. In the last step we determine $N_i$ by fitting Eq. 5.

An example for the results of this procedure is shown in Fig. 2. The example shows the FAP derived from ten thousand samples of Gaussian noise, where each sample has 60 data points and a sampling of 1 $\text{day}^{-1}$, thus, the complete time interval length is 60 days. The frequency range used for the analysis is $\Delta f = 1/2 - 1/60\ \text{day}^{-1}$. The frequency sampling during these simulations (and all other simulations) is fixed with respect to the length of the time interval, thus, the duration of observations $T$, and the frequency range. We evaluated the LSP at $N_{freq}$ equally spaced frequencies in the frequency range $\Delta f$, where $N_{freq} = 4T\Delta f$,

which was shown to be an adequate sampling to observe all possible peaks by Cumming et al. (1999). The blue coloured circles show the results for $\mathrm{Prob}(z > z_0)$ at a single frequency. The theoretical curve of Eq. 4 is shown in magenta. The determined probability and the theoretical one match very well. The results for the FAP (1-CDF) are shown as black coloured circles. The red curve is determined by fitting Eq. 5 to these data points. The number of independent frequencies $N_i$ in this case is about 72. From this curve different FAP levels can be determined. In the following we typically use a FAP level of 5%, which means that in only 5% of the noise samples the maximum peak in the complete frequency range exceeded the corresponding peak height value. In Fig. 2 the dashed horizontal line marks a FAP of 5% and the intersection with the red curve gives the height of about 0.225 that corresponds to this level.

## 3.2 Dependency of $N_i$ and FAP

The number of independent frequencies $N_i$ and the false alarm probability depends on different factors: the length of the analysed time interval $T$, the data gaps within the time interval and the analysed frequency range $\Delta f$. Since in the period analysis of the OH* temperatures different situations with respect to data gaps can occur and, additionally, the length of the window (time interval) and the frequency range can be chosen, one would have to perform simulations for all situations. As these simulations are much more time-consuming than the calculation of the LSP itself, we want to avoid these numerous simulations. Thus, we examined the different dependencies to find a faster and easier way to determine $N_i$ and, thus, the FAP levels. The sampling of the time series used for these analyses was chosen to be 1 day$^{-1}$, which is the same as for the nightly mean OH* temperatures without data gaps. For the different analyses we varied only one parameter and kept the other two fixed. In all cases ten thousand noise samples were used to determine one $N_i$ value.

Firstly, we analysed the dependency of $N_i$ on the length of the time interval $T$. Here the frequency range was kept constant and the time series had no data gaps. As this is the case and the sampling is 1 day$^{-1}$, the length of the time interval is equal to the number of data points $N$, i.e. a time interval of 60 days has 60 data points. The frequency range was fixed to $\Delta f = 1/2 - 1/60$ day$^{-1}$ for the first analysis. Since the width of a peak is inversely proportional to the length of the analysed time interval (see e.g. Cumming et al., 1999; Zechmeister and Kürster, 2009), the number of independent frequencies $N_i$ for a fixed frequency range should linearly increase with increasing time interval length. Figure 3 shows the results for $N_i$ for different time interval lengths $T$ between 30 and 90 days (typical values used for the analysis of nightly mean OH* temperatures) as blue full circles. Obviously, the dependency is linear. A linear fit including an additional intercept leads to an intercept of about zero. Thus, we calculated a fit line that has to intersect the point (0,0) and only determined the slope of this line, which is 1.208 ($\pm$ 0.004) days$^{-1}$. The fit is shown as a blue line in Fig. 3 a). Since the number of data points $N$ increases with increasing length of the time interval $T$, the probability that the power at a single frequency exceeds a certain value by chance decreases (compare Eq. 4). As this effect is larger than the opposite effect of the increase of $N_i$, the FAP levels also decrease. Fig. 3 b) shows the levels of a FAP of 5% for the different time interval lengths as blue full circles.

In a second analysis we varied the frequency range and repeated the analysis that was done before. The frequency ranges lay between $1/2 - 1/5$ day$^{-1}$ and $1/2 - 1/90$ day$^{-1}$. A smaller frequency range should include a smaller number of independent frequencies. As the decrease of $N_i$ for a reduction of $\Delta f$ depends on the width of the peaks, and therefore on the length

of the time interval $T$, the decrease of $N_i$ for the same reduction of $\Delta f$ has to be larger for larger $T$. This can be seen in Fig. 3 a), where example results for the frequency ranges $1/2 - 1/5$ day$^{-1}$ and $1/2 - 1/10$ day$^{-1}$ are shown in black and red, respectively. For the smallest frequency range the lowest values can be seen and the largest decrease of $N_i$ is observed for the longest time interval $T$. Because of this dependency of the decrease of $N_i$ on the time interval length, the fit lines are not shifted by a constant value, but the slopes of the fit lines change. Thus, the slopes depend on frequency range $\Delta f$. Fig. 3 c) shows the dependency of the slopes on the frequency range $\Delta f$. Obviously, for the analysed frequency ranges this dependency can be described by a straight line. A fit to the data leads to the results for the slope of 2.92 ($\pm$ 0.02) day days$^{-1}$ and for the intercept of -0.203 ($\pm$ 0.008) days$^{-1}$. The fit line is shown as black line. With the knowledge of these parameters the number of independent frequencies $N_i$ can be determined for each combination within the analysed parameter range by

$$N_i = (2.92 \text{ day days}^{-1} \cdot \Delta f - 0.203 \text{ days}^{-1}) \cdot T \qquad (6)$$

In the last analysis we evaluated the dependency of $N_i$ on the number of data gaps in a fixed time interval. The frequency ranges for this analysis were $\Delta f = 1/2 - 1/5$ day$^{-1}$, $1/2 - 1/10$ day$^{-1}$, and $1/2 - 1/60$ day$^{-1}$. We took a time interval of 60 days and introduced 1 to 29 randomly distributed data gaps. We only removed data points inside the complete time interval, i.e. both end points were always there and the time interval length was always 60 days. Since the spectral width of the peaks depends on the length of the time interval, which is fixed, and not on the number of data points, the number of independent frequencies $N_i$ is supposed to be almost the same for different numbers of data gaps. Fig. 4 a) shows $N_i$ in dependency of the number of data gaps for different frequency ranges. In all cases only a slight decrease of $N_i$ with increasing number of gaps can be seen. The decrease is slightly larger for that frequency ranges that lead to larger $N_i$ values. But the relative decrease is very similar for all shown situations. The decrease in all cases is only on the order of a few percent for 50% data gaps. This decrease is caused by an on average very small decrease of the resolution caused by a small increase of the peak width. Although the number of independent frequencies is nearly constant, this does not mean that the FAP levels stay the same. Since the number of data points $N$ decreases with increasing number of data gaps, the probability that the power at a single frequency exceeds a certain value increases (compare Eq. 4). Thus, the FAP for a certain peak height also increases. This increase is shown in Fig. 4 b). The effect of the decrease of $N_i$ on the FAP levels of 5% is typically on the order of a few ‰. Thus, a non-consideration of this decrease of $N_i$ would lead to a very small change of the FAP levels. Furthermore, the change when considering the decrease would be negative, i.e. the FAP for the same height $z$ would get smaller. Consequently, the FAP levels of 5% also have smaller values. Thus, a non-consideration would not change the judgement if a signal is significant or not in a false way. When a signal exceeds a higher value it will certainly exceed a smaller value, too. Nonetheless, in the FAP levels shown later on the effect of the data gaps on the $N_i$ values is considered.

## 4 Data evaluation

### 4.1 Artificial data

In order to study the performance of the approach we analysed different time series of artificial data. In this section we present selected examples of these time series. The total length of the time series was always 1 year (365 days) and the sampling was 1 day$^{-1}$, which is the same as for the nightly mean OH$^*$ temperatures without data gaps.

The analysis of a single sinusoid is a very trivial problem and the approach delivers the expected results (not shown). As the approach shall be used in the case of non stable periodicities, we focus here on such problems. The first example shows a time series of a periodic signal with a period that increases with time from approximately 8 days to 16 days and an amplitude of 1 K. The time series is shown in Fig. 5 a) as black curve (the components signal (blue) and noise (green) are shown additionally in seperate panels). The results of the analysis are shown in Fig. 5 b) and c) for the normalised power and the amplitude, respectively. The y-axes of these two figures give the frequency and period, respectively, and the x-axes show the center days of the sequentially analysed time intervals. The normalised power and the amplitude are shown color coded and the white contour lines mark the FAP level of 5% ($N_i$ was determined using Eq. 6). The results clearly show the change of the period with time and the normalised power is close to one. The small deviation from a value of one can be explained by the change of the period which occurs on a smaller time scale than the interval size of 60 days. Thus, a sinusoid with a fixed period is not able to explain the complete variance in each of the analysed time intervals. The results for the amplitude show values close to 1 K, and, thus, also the expectation. The analysis was repeated for the same periodic signal with additional noise added to the time series and also data gaps that have been incorporated. The standard deviation of the noise was 0.5 times the standard deviation of the signal and, thus, the variance of the noise is one quarter of that of the signal. Additionally, about 30% of the data points have been randomly removed. The signal with gaps (blue curve), the noise (green curve) and the the complete time series (sum of both; black curve) are shown in Fig. 5 d). The corresponding results are shown in Fig. 5 e) and f). The displayed FAP level of 5% was determined for each LSP individually with respect to the varying length of the time interval (when end points are missing) and the number of data points inside these time intervals. Additionally, the small decrease of $N_i$ due to the data gaps was considered (see Sect. 3.2). The change of the period is still captured very well. In the case of noise and data gaps the normalised power reduces to a value of about 0.8 as a part of the variance can be explained by the contribution of the noise (ratio 4 to 1 for signal to noise). The amplitude shows some fluctuations, but these fluctuations go around a value of 1 K. Additionally, the noisy behaviour at smaller periods is much better visible for the amplitudes compared to the powers, because the square root of the powers enters the calculation of the amplitudes (compare Fig. 3) and therefore differences to the maximum amplitude get smaller. In total, the results clearly capture the main features of the time series with respect to period, amplitude, and explained variance.

We additionally present two further examples. The time series and the results of the analyses are shown in Fig. 6. The first time series is composed of a periodic signal with a period of 25 days and an amplitude that varies between 0 and 1 K (Fig. 6 a) blue curve in upper panel) and additional noise (Fig. 6 a) green curve in middle panel). The standard deviation of the noise was again 0.5 times the standard deviation of the signal and about 30% of the data points have been removed. The complete time series

is shown as a black curve in the lower panel of Fig. 6 a). The results for the normalised power and the amplitude are shown in Fig. 6 b) and c), respectively. The normalised power shows an increasing value to the center of the complete time interval. This behaviour is caused by the contribution of the noise to the total time series, which is much larger when the amplitude is small and decreases with increasing amplitude of the signal. The result for the amplitude nicely reflects the increase of the amplitude to the center and the following decrease to the end of the time series. As the variation of the amplitude occurs on a smaller time scale than the chosen time interval for the analysis some kind of averaging occurs. Thus, the theoretical maximum of 1.0 K is not reached and the maximum value that is observed is about 0.9 K. In total, the main features of the signal are captured very well by the analysis and the correct period and the variation of the amplitude with time are detected. The last example shows the sum of the two former ones. Thus, the complete time series (Fig. 6 d) black curve in lower panel) is composed of a sinusoid with an amplitude of 1 K and an increasing period (Fig. 6 d) blue curve in upper panel), a periodic signal with a period of 25 days and an amplitude that varies between 0 and 1 K (Fig. 6 d) red curve in second panel), and noise (Fig. 6 d) green curve in third panel). The standard deviation of the noise and the amount of data gaps are the same as before. The results for the normalised power and amplitude are presented in Fig. 6 e) and f), respectively. The first signal can significantly be detected during the whole time and the increase of the period from about 8 to 16 days is captured very well. As the amplitude of the second signal increases to the center of the complete time interval, this signal can only be significantly detected in the middle of the complete time interval. The normalised power reflects the different contributions of the two signals to the complete time series very well. In the middle of the complete time series each single signal contributes to almost the same amount as the amplitude is about 1 K in both cases. The remaining part of the total variance can be explained by the noise (variance of noise is 0.25 times variance of sum of signals). At the beginning and the end nearly only the first signal and additionally the noise contribute to the complete time series. The results for the amplitude also show the main features of the two signals. For the first signal the amplitude stays at around 1 K during the whole time and the increasing and thereafter decreasing amplitude behaviour of the second signal is also captured. Compared to the former example the result for the amplitude is noisier, because of the larger absolute noise in the last example.

In summary, the applied method is able to detect periodic signals that vary with time, i.e. the amplitude or the period changes with time. In cases where changes occur on much smaller time scales than the used time window the results show some kind of averaging. Then the maximum values of the amplitude or the explained variance cannot be obtained and a mean value in the analysed time window is derived. The method is also very useful when noise is added to the time series and additionally data gaps are introduced. Although about 30% of the data points have been removed, the results are very good and still reflect the behaviour of the signals. Thus, the presented method is well suited to analyse time-varying periodicities even in the case of unequally spaced time series.

## 4.2 Measurement data

The $OH^*$ temperatures are derived from measurements by a GRIPS (GRound-based Infrared P-branch Spectrometer) instrument operated in Wuppertal (51° N, 7° E, Germany). This GRIPS instrument measures 3 emission lines of the $OH^*(3,1)$ band, the P1(2), P1(3) and P1(4) line. The relative intensities of these lines are used to derive rotational temperatures (Bittner et al.,

2000, and references therein). The OH layer from which the emissions originate is located in the mesopause region. The mean altitude is about 87 km and the layer has a full width at half maximum (FWHM) of about 9 km (e.g. Baker and Stair, 1998; Oberheide et al., 2006). Measurements are carried out every night, except for nights with bad weather conditions. The OH$^*$ temperatures have been continuously observed from Wuppertal since mid-1987 and a GRIPS instrument is still in operation to continue the observations. Until mid-2011 the measurements have been carried out by the GRIPS-II instrument (see Bittner et al., 2000, 2002, for an instrument description) and after then the GRIPS-N instrument (follow-up of GRIPS-II) is used to continue the observations (Kalicinsky et al., 2016).

Fig. 7 shows the nightly mean OH$^*$ temperatures for the year 1989 as an example. This year was chosen because Bittner et al. (2000) analysed the same year with a different technique (wavelet transform) and, thus, the results of our approach can be compared to their results. The temperatures show the typical seasonal behaviour with a temperature minimum in summer and a maximum in winter. This behaviour can be described with three main components: an annual, a semi-annual, and a ter-annual cycle (Bittner et al., 2000). The red curve in the figure shows a least squares fit to the data that considers these three components. Such fits are typically used to determine the annual average OH$^*$ temperatures since a simple arithmetic mean is not advisable because of the data gaps (e.g. Bittner et al., 2002; Offermann et al., 2010; Perminov et al., 2014; Kalicinsky et al., 2016). The lower panel of Fig. 7 shows the residual temperatures, i.e. the OH$^*$ temperatures minus the determined fit curve. Bittner et al. (2000) already showed that such residual temperatures include statistically significant periodic fluctuations. We now analyse the residual temperatures with respect to such fluctuations using the moving LSP approach.

The results for the normalised power and the amplitude are shown in Fig. 8. Different events with significant periodic fluctuations can be detected when using the moving LSP approach. The largest event is detected at the beginning of the year. The determined period is about 40 days and the amplitude 6 to 7 K. This behaviour can also be seen in the residual temperatures just by eye (compare Fig. 7). It seems that the fluctuations continue with a slightly larger period and smaller amplitude, but the result cannot be judged as significant after a center day of the interval of about 70 days. As can be seen in the residual temperatures the number of observations between day 75 and 125 is very low and a lot of data gaps are present. The FAP levels for time windows including a large number of data gaps increase then and, thus, the results are not significant, although it is likely that the signal is still there and real. Additionally, the data gaps are responsible for the vertical structure that can be clearly observed in the amplitudes in this time region, because the gaps interupt the continuity. Around a center day of 250 days a second significant result for a fluctuation with a period of about 50 days is detected, but the amplitude is smaller with 4 to 5 K. At the end of the year additional significant events with smaller periods of about 10 and 16 days can be seen. All of these significant fluctuations agree well with the findings of Bittner et al. (2000), where the authors analysed the same observations using a wavelet transform and assimilation technique based on the maximum entropy method to get rid of the data gaps. Our new method enables now a safe detection of such significant fluctuations without the need of processing the data before the analysis.

## 5 Summary and conclusions

We present an approach to analyse time series with unequal spacing with respect to significant period fluctuations. The approach is also able to derive time-resolved information on the periods and amplitudes of the detected fluctuations. It is based on the classical Lomb-Scargle periodogram (LSP), a method that can handle unequally spaced time series. Additionally, it uses the idea of a moving window to enable the determination of time-resolved periods and amplitudes. The significance of the results is analysed with the typically used false alarm probability (FAP). As the determination of the FAP levels needs many simulations, we derived the dependencies of the FAP levels on the length of the analysed time interval $T$, the frequency range $\Delta f$, and the number of data gaps to find a fast and easy way to calculate the FAP levels in the used parameter range. Thus, we can avoid a large number of simulations. In the analysed parameter range the number of independent frequencies $N_i$ shows a linear dependency on the length of the time interval $T$, because the peak width is inversely proportional to $T$. Furthermore, the slope of the line that describes this dependency is different for different frequency ranges, where a smaller frequency range $\Delta f$ reduces the slope. We used these two relationships to fastly calculate the FAP levels. The number of data gaps has only a very minor effect, because the peak width depends on the length of the time interval and not on the number of data points. The approach was tested with different artificially generated time series. These time series include variations of the period and amplitude with time and, additionally, noise is added and data gaps have been introduced. In all cases the approach shows very good results and, thus, the approach is a suitable method for the time-resolved detection of periodic fluctuations, even in the case of unequal spacing. Finally, we analysed the nightly mean OH$^*$ temperatures that have been observed from Wuppertal ($51°$ N, $7°$ E, Germany) in the year 1989. The results show several significant events with fluctuations that have periods in the range between 10 and 50 days and amplitudes between 3 and 7 K. These significant results agree very well with the results of a former study carried out by Bittner et al. (2000) without the need of processing the data before the analysis.

*Data availability.* The nightly mean OH$^*$ temperatures can be obtained by request to the corresponding author or to Peter Knieling (knieling@uni-wuppertal.de).

*Author contributions.* CK conceptualised the method. CK and RR performed the simulations and did the analyses under intensive discussion with RK. PK provided th OH$^*$ data. The manusript was written by CK with contributions from all coauthors.

*Competing interests.* The authors declare that no competing interests are present.

*Acknowledgements.* This work was funded by the German Federal Ministry of Education and Research (BMBF) within the ROMIC (Role Of the Middle atmosphere In Climate) project MALODY (Middle Atmosphere LOng period DYnamics) under Grant no. 01LG1207A.

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

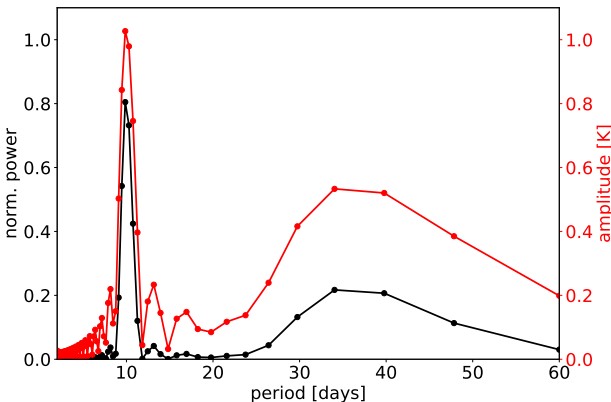

**Figure 1.** Example LSP for a time series composed of two sinusoids. The first one has a period of 10 days and an amplitue of 1 K and the second has a period of 35 days and an amplitue of 0.5 K. The normalised power is shown as black curve and the amplitude as red curve with a second axis to the right.

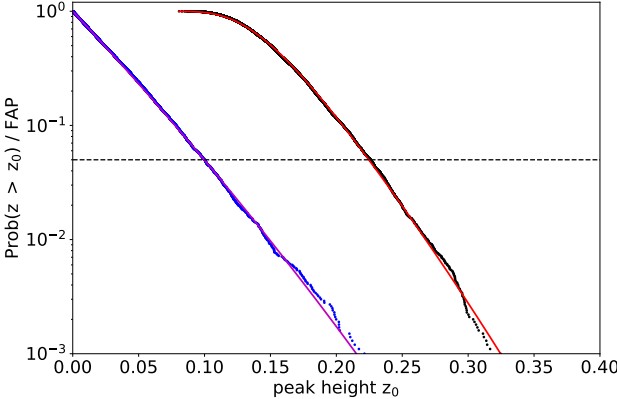

**Figure 2.** False alarm probability (FAP) and $\mathrm{Prob}(z > z_0)$ at a single frequency derived from ten thousand of noise samples with 60 data points each. The data sampling was $1\,\mathrm{day}^{-1}$ and the analysed frequency range $\Delta f = 1/2 - 1/60\,\mathrm{day}^{-1}$. The derived $\mathrm{Prob}(z > z_0)$ is shown with blue coloured circles and the theoretical curve (Eq. 4) is depicted in magenta. The determined FAP is shown by the black coloured circles and the fit to these data points using Eq. 5 is displayed as red curve. The dashed horizontal line marks a FAP of 5%.

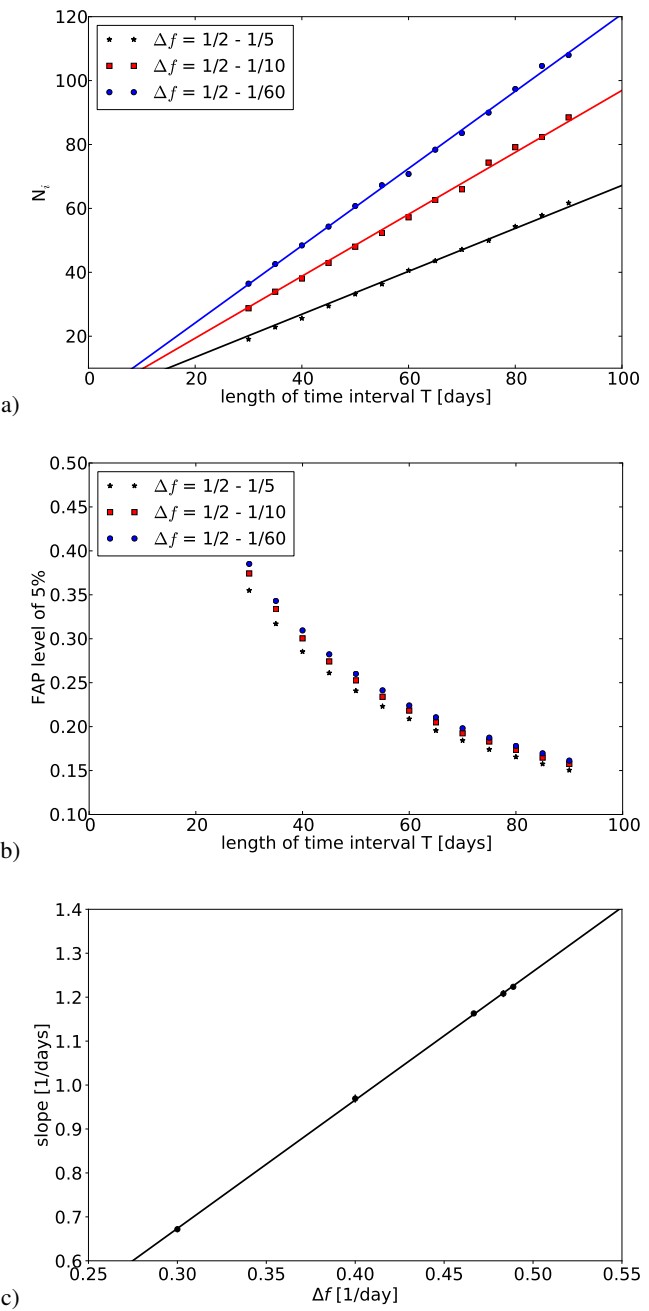

a)

b)

c)

**Figure 3.** a) and b): Dependency of $N_i$ and the FAP level of 5% on the length of the time interval $T$ and the frequency range. The analysed frequency ranges are $1/2 - 1/5 \ \mathrm{day}^{-1}$, $1/2 - 1/10 \ \mathrm{day}^{-1}$, and $1/2 - 1/60 \ \mathrm{day}^{-1}$ and the time series of the simulations have no data gaps. c): Dependency of the slopes (lines from panel a) and additionally for the frequency ranges $1/2 - 1/30 \ \mathrm{day}^{-1}$ and $1/2 - 1/90 \ \mathrm{day}^{-1}$) on the frequency range $\Delta f$. The error bars show two times the standard error of the slopes.

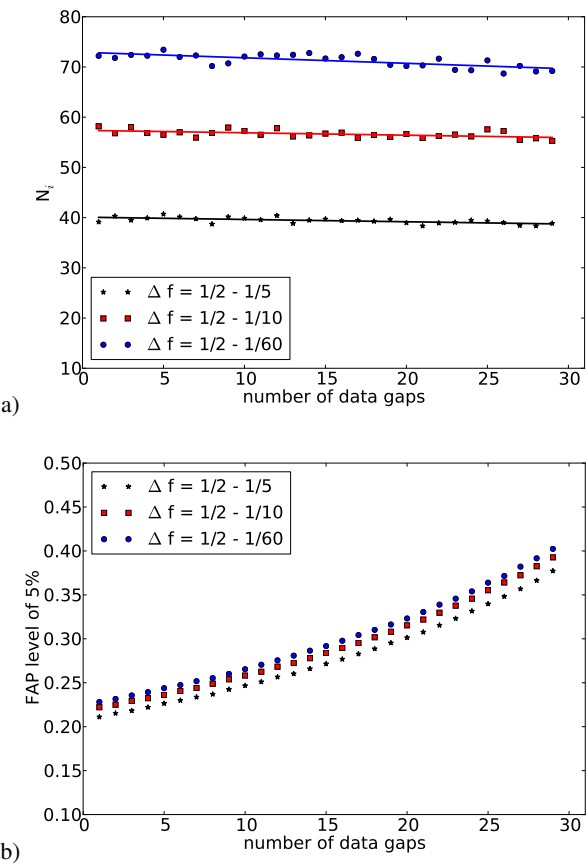

a)

b)

**Figure 4.** Dependency of $N_i$ and the FAP level of 5% on the number of data gaps in a fixed time interval of length $T$. The analysed frequency ranges are $1/2 - 1/5 \ \mathrm{day}^{-1}$, $1/2 - 1/10 \ \mathrm{day}^{-1}$, and $1/2 - 1/60 \ \mathrm{day}^{-1}$ and the time interval length is 60 days.

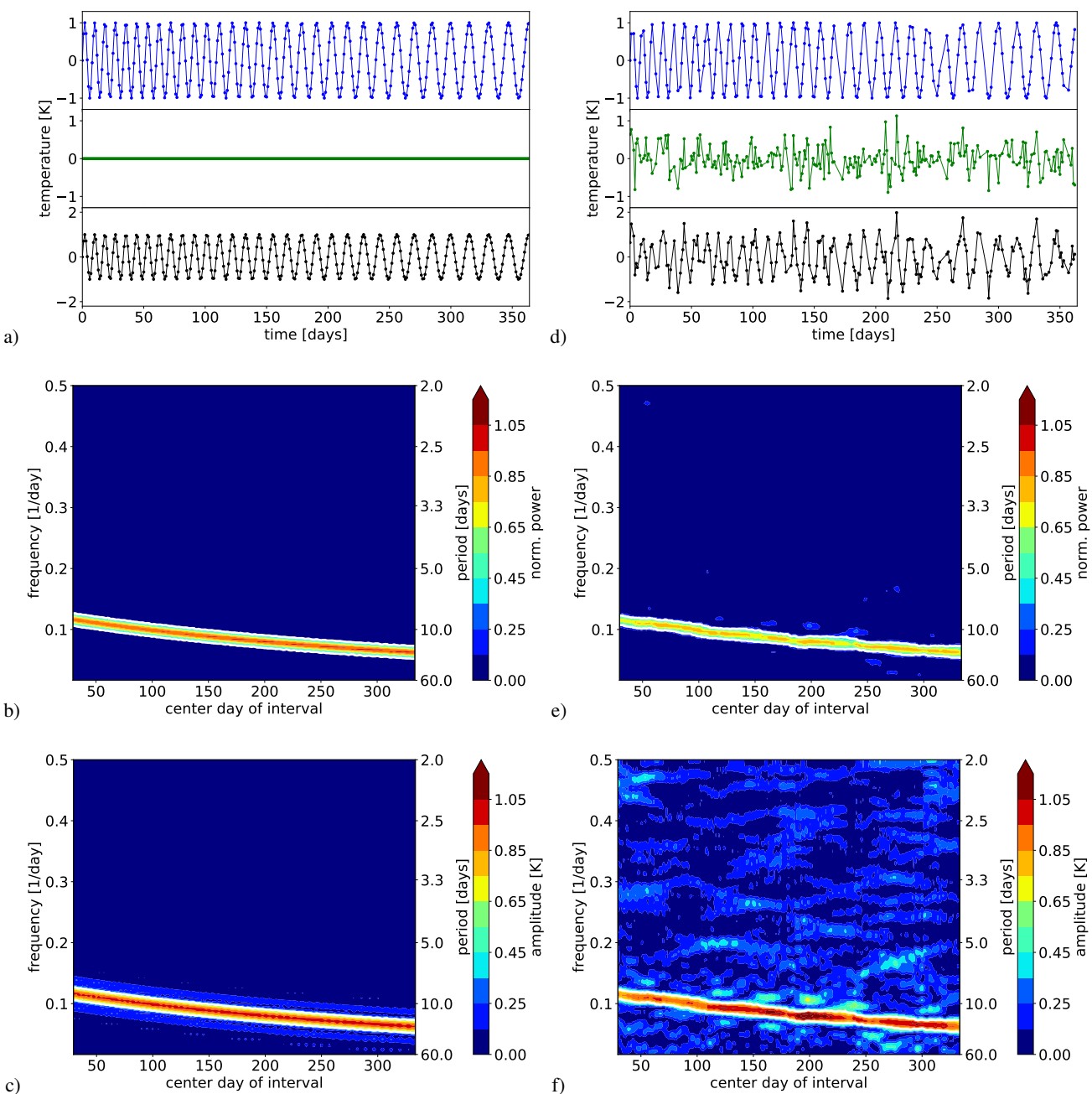

**Figure 5.** a): Time series of a periodic signal with increasing period. The upper panel shows the signal, the middle panel the noise and the lower panel the sum of both. b) and c): Results for the normalised power and amplitude. The results are displayed at the center day of the corresponding time window. The length of the time window was 60 days. The white contours mark the significant results. d) – f): Same as for a) – c) with additional noise added to the time series and data gaps.

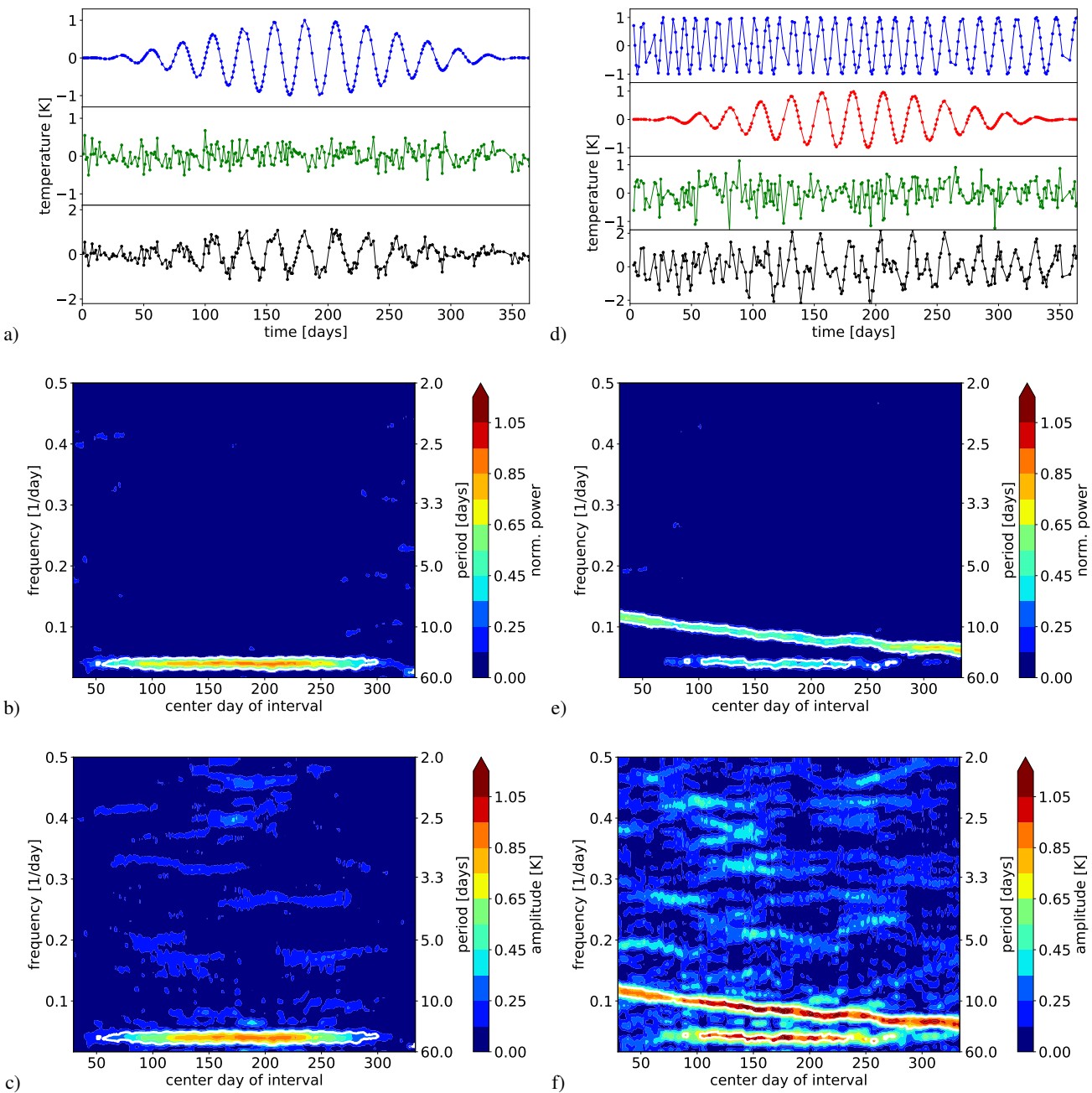

**Figure 6.** a): Time series of a periodic signal with varying amplitude and additional noise and data gaps. The upper panel shows the signal, the middle panel the noise and the lower panel the sum of both. b) and c): Results for the normalised power and amplitude. The results are displayed at the center day of the corresponding time window. The length of the time window was 60 days. The white contours mark the significant results. d): Time series of a periodic signal with increasing amplitude plus a periodic signal with varying amplitude and additional noise and data gaps. The upper two panels show the two signals, the third panel the noise and the lower panel the sum of all. e) and f): Same as for b) and c).

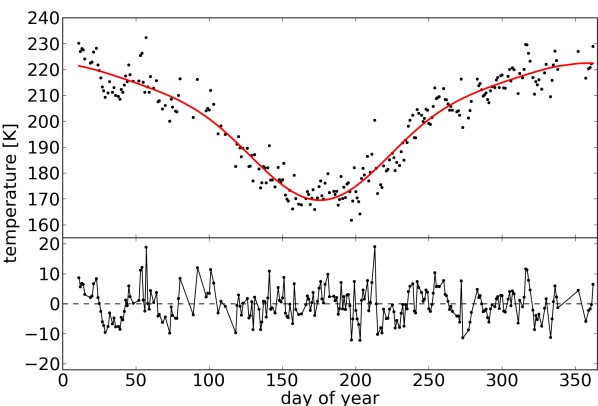

**Figure 7.** Nightly mean OH$^*$ temperatures observed from Wuppertal in the year 1989. The red curve shows the fit of the seasonal cycle including an annual, semi-annual, and ter-annual component. The residual temperatures (measurements minus fit) are shown in the lower panel.

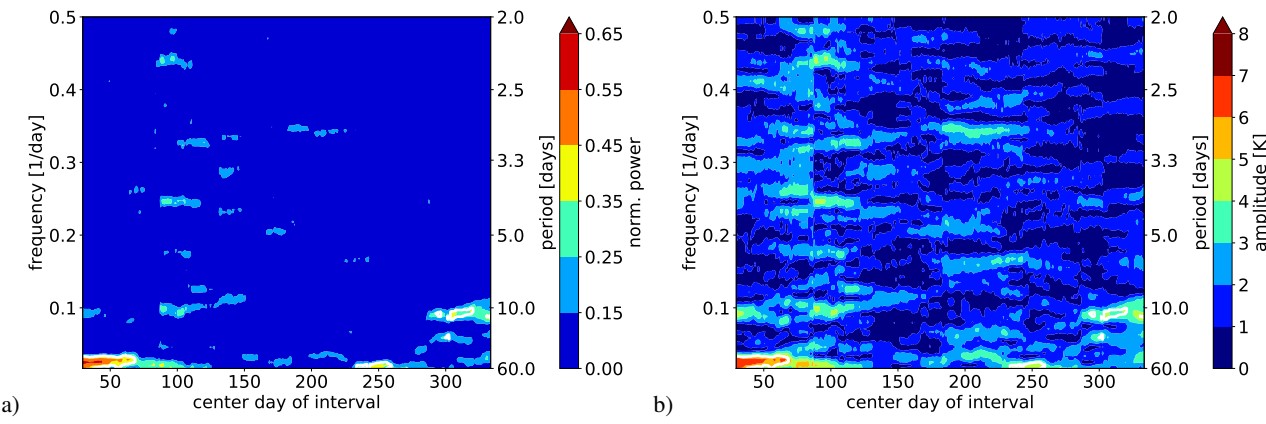

**Figure 8.** Results for the normalised power and amplitude for the analysis of the temperature residual of the GRIPS observations in 1989. The results are displayed at the center day of the corresponding time window. The length of the time window was 60 days. The white contours mark the significant results.