# Peer review of "Determination of time-varying periodicities in unequally spaced time series of OH* temperatures using a moving Lomb-Scargle Periodogram and a fast calculation of the false alarm probabilities"

_Atmospheric Measurement Techniques, 2019_

## Referee Comment (RC1) · Anonymous Referee #1 · 30 Oct 2019

Report on manuscript AMT-2019-332 Moving Lomb-Scargle Periodogram: A way to identify time-varying periodicities in unequally spaced time series of OH* temperatures, by Christoph Kalicinsky et al.

This paper presents the application of the LS periodogram analysis, ("Moving" LS periodogram) as a method to identify variations in unequally spaced OH temperature time series, and the calculation of the FAP as a method to test the significance of the peaks obtained in the LS periodogram analysis. However the use of "moving" periodogram is not new in studies of airglow variability in general, neither in OH airglow variability in particular. Then, although the authors describe the necessary mathematical method-

ology and later they apply it to a particular set of OH airglow data the authors forget to discuss many airglow studies that have previously used this mathematical tool for variability analysis. Then I can not recommend this work for publication until this is resolved.

General comments:

As the authors comment in the manuscript, airglow data have many gaps (day-night periods, moon-periods, weather...) and airglow variations are not constant, there are some features "more stable, repetitive and stronger" and other "more unstable and smaller...". There are different papers (quite a few) that have been devoted to study airglow variability (at different temporal scales, gravity waves, planetary waves, tides, seasonal variations...) by using time series analysis sliding a temporal window over a data set. This should be clarified and mentioned in the text.

Moreover, airglow studies dealing with periodogram calculations have also needed a method to distinguish the significant peaks, it is to say (real) peaks from the noise and in this sense, levels of confidence that a peak be a signal have been used in airglow studies that should also be mentioned (signals well above of the noise level, probability that a peak (z) above a level (z0) be false, false alarm probability function, $FAP=1-(1-Prob(z>z0)^{Ni}$ or confidence level...). However, as the authors point out, one of the difficulty to evaluate the FAP is to find the number of independent frequencies "Ni" of the data due to the non-orthogonality between different frequencies, by that "Ni" is usually calculated by fitting the FAP equation using different bootstrap simulations of the data set.

In the present manuscript the authors analysis the number of independent frequencies in different samples to find an analytic expression for this parameter (Ni). They find that Ni increases linearly with the length of interval (T), $Ni=slope(f\_rang) \times T$, but the slope for each frequency range analysed ($f\_rang$) also follow a linear relation with the range of frequencies ($slope=m(f)+b$), obtaining an analytic expression for Ni, $Ni= (m(f)+b) \times$

T. Finally in section 4.2, they use this relationship to find the number of independent frequencies of a set of OH measurements to easily calculate the FAP at one level.

I think this paper may be accepted for publications, but although the paper properly presents the necessary mathematical tools, and the enough set of simulations to find a mathematical expression to easy calculate the number of possible independent frequencies necessary to evaluate the FAP, the paper does not mention some of the works that have been carried out in the studies of airglow variability by using periodogram analysis + moving window + significance test (including FAP). In this sense:

1)The title of the paper should change because it seems that a "new" method "moving periodogram" is "first" applied in OH airglow studies, and that is not true.

2)The introduction should be improved to give appropriate credit to previous work in airglow variability studies using periodogram analysis.

---

## Referee Comment (RC2) · Anonymous Referee #2 · 23 Nov 2019

General comments:

This manuscript describes an analysis technique to provide spectral and temporal information based on time series with unequal spacing. It is based on a windowed Lomb-Scargle periodogram analysis. The technique is kind of the analogue of a wavelet transform for unequally spaced data. It is of importance for data analysis in many different fields. The technique is certainly not only applicable to OH rotational temperature time series and I suggest removing this emphasis from the paper (particularly from the first sentence of the abstract). The paper also leaves it open, whether this technique is frequently applied in other fields. If this is a well-established technique and the main

point of the paper is that it is applied to OH temperature measurements for the first time, this should be explicitly stated. Conversely, if this is not a routine technique, this should be mentioned as well. The paper is well and carefully written, in my opinion. I ask the authors to consider the general comments above and the specific comments below and recommend accepting the paper subject to minor revisions.

Specific comments:

Line 1: "We present an approach to analyse time series of OH temperatures with unequal spacing"

The approach is applicable to all kinds of unequally spaced time series, right? Why narrow its applicability down to OH rotational temperatures?

Line 17: "are useful at all."

Do you mean "are not useful at all" or "are useful" ?

Lines 24 and 26: "wavelet transformation" -> "wavelet transform"

Line 37: "The power is defined as"

This is "spectral power", right? Perhaps "power" can be specified further.

Lines 80 and following: Is this sample time series equally spaced? This should perhaps be mentioned.

Line 86: "amplitudes itselves" -> "amplitudes themselves"

Line 101: "independent frequencies"

I suggest providing a brief qualitative statement as to what "independent frequencies" means in this context. Most readers will perhaps guess the correct meaning, but it would be good to define the term.

Line 107: "From this maximum peak heights" -> "this .. height" or "these .. heights" ?

Line 108: "Consequently, the FAP is then 1-CDF."

This cannot be derived from the information provided in this paper, right? I suggest giving a reference for this.

Line 155: I suggest to replace "line" by "straight line"

Same line: only the numerical values of the slope and intersect are given. I suggest providing the units as well.

Equation (6): please provide the units of the quantities.

Line 162: "Since the peak width"

I suggest mentioning explicitly that "width" refers to the "spectral width"

Line 174: "levels .. has" -> "levels .. have"

Line 191: "deviation to" -> "deviation from"

Fig 5: The black lines in panels b, c, e, f are difficult to see in some panels -> perhaps white lines? If yes, then this should also be changed in the rest of the figures.

Line 213: "As the variation of the amplitude occurs on a smaller time scale than the chosen time interval for the analysis the maximum value reached is about 0.9 K."

I read this sentence several times, but I don't really understand the argument. Can it be expressed in a better way?

Figures 5f, 6f, and 8b: A brief comment on the signatures at shorter periods would be appropriate? They are not significant, but they stick out. Are these some kind of harmonics?

Section 4.2: Perhaps you can mention briefly, why the 1989 was chosen?

Line 268: "wavelet transformation" -> "wavelet transform"

Figure 8b: around day 85 there appears a "vertical structure". What is it caused by?

Probably gaps in the data. I suggest adding a brief comment to the paper.

---

## Author Comment (AC1) · 18 Dec 2019

Reply to the comments by reviewer 1 on the manuscript

"Moving Lomb-Scargle Periodogram: A way to identify time-varying periodicities in unequally spaced time series of OH* temperatures"

by C.Kalicinsky et al.

We thank the reviewer for his helpful comments and recommendations. In the following, we discuss the issues addressed by the reviewer and explain our opinions and the modifications of our manuscript.

We enumerate the comments and repeat them in bold face. The modifications of the manuscript are displayed in the marked-up manuscript version as colored text. Deleted parts are shown in red and new or modified text parts in blue.

**1 Comments Reviewer 1**

**This paper presents the application of the LS periodogram analysis, ("Moving" LS periodogram) as a method to identify variations in unequally spaced OH temperature time series, and the calculation of the FAP as a method to test the significance of the peaks obtained in the LS periodogram analysis. However the use of "moving" periodogram is not new in studies of airglow variability in general, neither in OH airglow variability in particular. Then, although the authors describe the necessary mathematical methodology and later they apply it to a particular set of OH airglow data the authors forget to discuss many airglow studies that have previously used this mathematical tool for variability analysis. Then I can not recommend this work for publication until this is resolved.**

**General comments:**
**As the authors comment in the manuscript, airglow data have many gaps (day-night periods, moon-periods, weather...) and airglow variations are not constant, there are some features "more stable, repetitive and stronger" and other "more unstable and smaller...". There are different papers (quite a few) that have been devoted to study airglow variability (at different temporal scales, gravity waves, planetary waves, tides,seasonal variations...) by using time series analysis sliding a temporal window over a data set. This should be clarified and mentioned in the text.**
**Moreover, airglow studies dealing with periodogram calculations have also needed a method to distinguish the significant peaks, it is to say (real) peaks from the noise and in this sense, levels of confidence that a peak be a signal have been used in airglow studies that should also be mentioned (signals well above of the noise level, probability that a peak (z) above a level (z0) be false, false alarm probability function, FAP=1-(1-Prob(z>z0)^Ni or confidence level...). However, as the authors point out, one of the difficulty to evaluate the FAP is to find the number of independent frequencies "Ni" of the data due to the non-orthogonality between different frequencies, by that "Ni" is usually calculated by fitting the FAP equation using different bootstrap simulations of the data set.**

In the present manuscript the authors analysis the number of independent frequencies in different samples to find an analytic expression for this parameter (Ni). They find that Ni increases linearly with the length of interval (T), Ni=slope(f_rang) x T, but the slope for each frequency range analysed (f_rang) also follow a linear relation with the range of frequencies (slope=m(f)+b), obtaining an analytic expression for Ni, Ni= (m(f)+b) xT. Finally in section 4.2, they use this relationship to find the number of independent frequencies of a set of OH measurements to easily calculate the FAP at one level.

I think this paper may be accepted for publications, but although the paper properly presents the necessary mathematical tools, and the enough set of simulations to find a mathematical expression to easy calculate the number of possible independent frequencies necessary to evaluate the FAP, the paper does not mention some of the works that have been carried out in the studies of airglow variability by using periodogram analysis + moving window + significance test (including FAP). In this sense:

1. **The title of the paper should change because it seems that a "new" method "moving periodogram" is "first" applied in OH airglow studies, and that is not true**

   We changed the title to put more emphasis on the OH time series analysis and, additionally, included the fast calculation of the FAP levels as this point is important for the easy application of the method. The new title is:
   Determination of time-varying periodicities in unequally spaced time series of OH* temperatures using a moving Lomb-Scargle Periodogram and a fast calculation of the false alarm probabilities

2. **The introduction should be improved to give appropriate credit to previous work in airglow variability studies using periodogram analysis.**

   We again searched the literature of OH and airglow studies dealing with all kind of variations from gravity waves to seasonal variations. We found several studies using the LSP for time series analysis but without a moving window approach. Only a few studies showing LSP for independent time periods following each other were found. We observed more studies using the wavelet transform after interpolation of data gaps, even very recent ones. This, in our opinion additionally motivates our approach. Thus, we do not believe that a moving LSP is a common or well-established approach in studies of OH analysis.
   We additionally expanded our search to other fields dealing with variations in the mesosphere and lower thermosphere region such as radar observations of winds. Here we found studies using LSP or other periodograms with moving windows, but either the significance evaluation was missing or the moving steps were much larger than the minimum possible ones. But this shows, as the reviewer mentioned, that a moving periodogram is surely not completely new.
   However, we think that our approach, especially when considering the fast and easy calculation of the FAP levels, is beyond that techniques frequently used in the field of OH analysis. Nonetheless, we surely revised the introduction and included all of these information to other studies to give credit to the other authors.

---

## Author Comment (AC2) · 18 Dec 2019

Reply to the comments by reviewer 2 on the manuscript

"Moving Lomb-Scargle Periodogram: A way to identify time-varying periodicities in unequally spaced time series of OH* temperatures"

by C.Kalicinsky et al.

We thank the reviewer for his helpful comments and recommendations. In the following, we discuss the issues addressed by the reviewer and explain our opinions and the modifications of our manuscript.
We enumerate the comments and repeat them in bold face. The modifications of the manuscript are displayed in the marked-up manuscript version as colored text. Deleted parts are shown in red and new or modified text parts in blue.

**1 Comments Reviewer 2**

**General comments:**
**This manuscript describes an analysis technique to provide spectral and temporal information based on time series with unequal spacing. It is based on a windowed Lomb-Scargle periodogram analysis. The technique is kind of the analoque of a wavelet transform for unequally spaced data. It is of importance for data analysis in many different fields. The technique is certainly not only applicable to OH rotational temperature time series and I suggest removing this emphasis from the paper (particularly from the first sentence of the abstract). The paper also leaves it open, whether this technique is frequently applied in other fields. If this is a well-established technique and the main point of the paper is that it is applied to OH temperature measurements for the first time, this should be explicitly stated. Conversely, if this is not a routine technique, this should be mentioned as well. The paper is well and carefully written, in my opinion. I ask the authors to consider the general comments above and the specific comments below and recommend accepting the paper subject to minor revisions.**

We again searched the literature of OH and airglow studies dealing with all kind of variations from gravity waves to seasonal variations. We found several studies using the LSP for time series analysis but without a moving window approach. Only a few studies showing LSP for independent time periods following each other were found. We additionally expanded our search to other fields dealing with variations in the mesosphere and lower thermosphere region such as radar observations of winds. Here we found studies using LSP or other periodograms with moving windows, but either the significance evaluation was missing or the moving steps were much larger than the minimum possible ones. Thus, we think that our approach, especially when considering the fast and easy calculation of the FAP levels, is beyond that techniques frequently used in the field of OH analysis (and maybe other fields). Nonetheless, we revised the introduction and included all of these information to other studies.

1. **Line 1: "We present an approach to analyse time series of OH temperatures with unequal spacing"**

**The approach is applicable to all kinds of unequally spaced time series, right? Why narrow its applicability down to OH rotational temperatures?**

This is correct. We removed "OH temperatures" from the sentence.

2. **Line 17: "are useful at all."**
**Do you mean "are not useful at all" or "are useful" ?**

We mean that the measurements are useful. We corrected this.

3. **Lines 24 and 26: "wavelet transformation" -> "wavelet transform"**

We changed this.

4. **Line 37: "The power is defined as"**
**This is "spectral power", right? Perhaps "power" can be specified further.**

Equation (1) gives the definition of the periodogram. The single values that are calculated at single frequencies then are periodogram (spectral) powers. We corrected the sentence and a few other points to make this clearer.

5. **Lines 80 and following: Is this sample time series equally spaced? This should perhaps be mentioned.**

This sample is equally spaced. We added this information.

6. **Line 86: "amplitudes itselves" -> "amplitudes themselves"**

We corrected this.

7. **Line 101: "independent frequencies"**
**I suggest providing a brief qualitative statement as to what "independent frequencies" means in this context. Most readers will perhaps guess the correct meaning, but it would be good to define the term.**

We added additional information on the idea behind the FAP and the different probabilities. The probability that a peak at a single frequency exeeds a certain value is $Prob(z>z0)$. Then $(1-Prob(z>z0))$ gives the probability that the peak is equal or below the certain value. In a frequency range then $(1-Prob(z>z0))^{N_i}$ gives the probability that all peaks are equal or below a certain value (this is given by the CDF of the maximum peaks; see 9.). Consequently, the FAP is then 1-CDF, thus $1-(1-Prob(z>z0)^{N_i}$. The number of independent frequencies is the number of frequencies where potentially peaks can occur.

8. **Line 107: "From this maximum peak heights" -> "this .. height" or "these .. heights" ?**

Here these heights is correct.

9. **Line 108: "Consequently, the FAP is then 1-CDF."**
**This cannot be derived from the information provided in this paper, right? I suggest giving a reference for this.**

As we added now some information to the derivation of the FAP (see 7.), we think this can be derived from the given information in the paper. However, we also added additional information in this paragraph.

10. **Line 155: I suggest to replace "line" by "straight line"**

We corrected this.

11. **Same line: only the numerical values of the slope and intersect are given. I suggest providing the units as well.**
**Equation (6): please provide the units of the quantities.**

We provided the units of the two given slopes, which is 1.208 $days^{-1}$ and the second slope 2.92 is $day/days$, as the length of interval is given in $days$ and frequency range in $day^{-1}$, we distinguish between these two. The dimension of the intercept is again $days^{-1}$.

12. **Line 162: "Since the peak width"**
**I suggest mentioning explicitly that "width" refers to the "spectral width"**

We added "spectral".

13. **Line 174: "levels .. has" -> "levels .. have"**

We corrected this.

14. **Line 191: "deviation to" -> "deviation from"**

We corrected this.

15. **Fig 5: The black lines in panels b, c, e, f are difficult to see in some panels -> perhaps white lines? If yes, then this should also be changed in the rest of the figures.**

We changed the black lines to white lines in all figures.

16. **Line 213: "As the variation of the amplitude occurs on a smaller time scale than the chosen time interval for the analysis the maximum value reached is about 0.9 K."**
**I read this sentence several times, but I don't really understand the argument. Can it be expressed in a better way?**

As the time interval of the analysis is smaller then the time scale of the variation some kind of averaging occurs. Thus, the maximum of 1 K is not reached and the observed maximum is about 0.9 K.

17. **Figures 5f, 6f, and 8b: A brief comment on the signatures at shorter periods would be appropriate? They are not significant, but they stick out. Are these some kind of harmonics?**

The signatures at shorter periods in Figures 5f, 6f, and 8b are mainly caused by noise. They appear much larger in the Figures of the amplitudes because there the square root of the power enters and, thus, the differences between these regions and the maxima get much smaller. We added a sentence to clearify this point.

18. **Section 4.2: Perhaps you can mention briefly, why the 1989 was chosen?**

This year was chosen because in the publication by Bittner et al. (2000) the same year was analysed with a different method and, thus, the results can be compared.

19. **Line 268: "wavelet transformation" -> "wavelet transform"**

    We corrected this.

20. **Figure 8b: around day 85 there appears a "vertical structure". What is it caused by? Probably gaps in the data. I suggest adding a brief comment to the paper.**

    The vertical structure is caused by the data gaps. We added a brief comment.